# Force–Displacement Hysteresis Model of Exterior PCRB Joints under Low Cyclic Loading

**DOI:** 10.3390/polym15092008

**Published:** 2023-04-24

**Authors:** Ping Wu, Yucong Guan, Feng Yu, Zilong Li, Yuan Fang

**Affiliations:** 1Department of Civil Engineering and Architecture, Anhui University of Technology, Ma’anshan 243032, China; 2Wuhu Technology and Innovation Research Institute, Anhui University of Technology, Wuhu 241003, China; 3Shanghai Baoye Group Corp., Ltd., Shanghai 200941, China

**Keywords:** FRP, PVC–FRP confined concrete, joint, column, beam, seismic behavior, skeleton curve, hysteresis loop

## Abstract

The seismic behaviors of exterior polyvinyl chloride–carbon-fiber-reinforced polymer (PVC–CFRP) confined concrete (PCCC) column-ring-beam joints (hereafter referred to as exterior PCRB joints) under low cyclic loading were investigated. A total of 11 specimens were tested to analyze the effects of the structural parameters (i.e., the reinforcement ratio of the ring beam, the CFRP-strip spacing, the ring-beam width, the reinforcement ratio of the frame beam, and the axial compression ratio) on the failure modes and hysteretic behavior. Three different failure modes, including the failure of the frame beam, the failure of the junction between the frame beam and the ring beam, and the shear failure of the ring-beam joint, were observed. The experimental results showed that the pinching effects of the hysteresis curves decreased and that the slope of the descending stage of the skeleton curves gradually decreased with the enhancement of the reinforcement ratio of the ring beam, axial compression ratio, and ring-beam width. The effects of the CFRP-strip spacing and the reinforcement ratio of the frame beam on the hysteresis loops and skeleton curves were marginal, while the reinforcement ratio of the frame beam exerted significant effects on the failure modes. Therefore, a model for predicting the skeleton curves of exterior PCRB joints and hysteresis rules was proposed, based on the softening constitutive relation of the concrete and a regression analysis of the test data. Ultimately, a load–displacement hysteresis model of the exterior PCRB joints was established and validated by the test data, with good agreement.

## 1. Introduction

Reinforced concrete (RC) column–beam-joint core areas are prone to brittle shear failure when subjected to axial force, bending moment, and shear force, which may induce the collapse of entire frame structures [1,2,3]. Therefore, as one of the most important components in RC-frame structures, the seismic behavior of the joint has been widely studied since the last century, and many substantial research results have been achieved [4,5,6,7,8,9]. However, due to the development of buildings of increasingly large size and height, and with irregular structures, traditional RC structures have been gradually replaced by new composite structures. For example, concrete-filled steel tubular (CFST) columns have been extensively studied and applied in engineering practices in recent years due to their high bearing capacity, convenient construction, and excellent ductility [10,11,12,13].

Traditional RC columns are usually connected to RC beams to form frame systems, while CFST columns can be connected to both RC beams and steel beams. Hence, many emerging joint connections between CFST columns and beams have been proposed and studied. Han and Li systematically investigated the seismic behaviors of CFST columns attached to steel-beam joints with RC slabs [14,15,16,17]. A low-cyclic-loading test was carried out on six specimens with two exterior joints and four interior joints, the hysteresis loops, skeleton curves, stiffness degradation, and ductility were analyzed, and it was found that these joint specimens exhibited good seismic performances [14]. In addition, a validation finite element model for composite joints was developed for parametric studies, and a hysteretic model for the panel zones of composite joints was proposed [15,16]. Xu et al. investigated the seismic behaviors of a flat CFST column attached to steel-beam joints through cyclic-loading tests of two exterior joints and two interior joints [18]. The experimental results illustrated that all the specimens showed ductile-beam-hinge failure and exhibited sufficient deformation capacity.

However, unlike connections to steel beams, connections between CFST columns and RC beams usually require the disconnection of the outer steel tube at the joint position to ensure that the RC beam’s longitudinal reinforcement passes through the joint-core area. This disconnection mode is conducive to the transfer of shear force and bending moment, improves the bonding performance of the longitudinal reinforcement and concrete in the joint core, and reduces the axial bearing capacity and stiffness of the joint core. To solve this problem, Fang et al. attempted to improve the bearing capacity of the joint core by increasing the cross-sectional area of the concrete in the joint-core area and allocating horizontal-ring reinforcement simultaneously; they demonstrated that this new ring-beam joint showed high axial bearing capacity [19]. Subsequently, Chen et al. designed cyclic-loading tests for six CFST-column–RC-ring-beam joints [20]. The hysteretic response, strain, and strength were discussed, and the favorable hysteretic performances and energy-consumption capacity of the connection were confirmed. Furthermore, Zhang et al. introduced an octagonal ring-beam-joint connection system, in which a steel cage was anchored inside the joint zone and an octagonal ring beam was located outside the CFST column [21]. Four column–beam composite joint specimens were tested under cyclic loading; the joints showed favorable seismic performances, and the anti-seismic design principles of “strong joints and weak members” were easily achieved. As can be seen from this overview, the joint connections between CFST columns and beams have been extensively studied and considerable achievements have been secured.

Similarly to CFST structures, as another novel composite structure, PVC–CFRP confined concrete (PCCC) columns have received increasing attention in research in the past two decades, owing to their high bearing capacity, light weight, strong corrosion resistance, and convenient construction [22,23,24,25,26,27,28]. Similarly, the connection between PCCC columns and RC beams is also a significant area of research focus. Referring to CFST columns, Yu et al. conducted experimental investigations on PCCC columns with ring-beam joints under axial loading, and models for predicting the ultimate axial strength and stress–strain relations were proposed [29,30,31]. Subsequently, Yu et al. performed a further experimental study and a finite element analysis of a PCRB joint subjected to eccentric compression and advanced some appropriate design suggestions [32].

To the best of our knowledge, few studies focus on the seismic behaviors of PCRB joints. However, as described above, the brittle shear failure of the joint-core area may induce the collapse of entire frame structures. Therefore, studies cannot be limited to static experiments if PCRB joints are to be applied in practical engineering. In this study, 11 exterior PCRB joints were tested to investigate the effects of the structural parameters on the seismic behaviors. Additionally, a P−Δ hysteresis model of the exterior PCRB joint was established. This model provides a preliminary theoretical guideline for elastic–plastic seismic-response analysis.

## 2. Experimental Program

### 2.1. Test Specimens and Material Properties

To systematically and comprehensively investigate the failure modes and seismic behaviors of the exterior PCRB joints, a total of 11 one-third-scale specimens were tested under low cyclic loading considering the general sizes of the specimens for engineering and the feasibility of the tests, in accordance with the provisions of Chinese code JGJ/T 101-2015 [33]. A exterior PCRB joint was composed of two column foundations, two PCCC columns, an enlarged ring-beam joint, and a RC-frame beam, as shown in Figure 1. Additionally, 5 structural parameters, including reinforcement ratio of ring beam $\rho_{r}$, reinforcement ratio of frame beam $\rho_{b}$, CFRP-strip spacing $s_{f}$, axial compression ratio *n* , and ring-beam width *b* were considered. The details of structural designs and the specific definitions of these different structural parameters are listed in Table 1.

The thickness and outer diameter of all PVC tubes were 7.8 mm and 200 mm, respectively. The width of CFRP strips was 20 mm. The 20-mm-wide CFRP strips with 2 layers were wrapped around the PVC tubes, and the thickness of each layer was 0.111 mm. Furthermore, 3 layers of CFRP strip were adopted in the junction between the columns and ring-beam joint to prevent the early destruction of columns end. Additionally, the strength grade of all concretes was C30. The ultimate strength and elastic modulus of concrete were obtained from the standard test of concrete cubes (150 mm × 150 mm × 150 mm). The measured properties of the all the materials (i.e., CFRP strips, PVC tube, concrete and reinforcement) used in the tests are summarized in Table 2.

### 2.2. Test Setup

Figure 2 shows the general view of the test setup of the exterior PCRB joint under low cyclic loading. A 2000-kN hydraulic jack was utilized to apply the constant axial force on the top of column. Meanwhile, cyclic loading was applied to the end of frame beam by the MTS hydraulic actuator (with a maximum capacity of 500 kN). The displacement-control loading scheme was adopted in the tests, and the detailed loading history is illustrated in Figure 3. Additionally, the test was terminated when the load decreased to 85% of the peak bearing capacity. Both the load and displacement at the frame-beam end were automatically collected by the data-acquisition system of MTS device.

## 3. Experimental Results and Discussion

### 3.1. Failure Modes

The failure process of all the specimens can be summarized as three typical stages (i.e., elastic, crack development, and failure). Initially, the deformation of all the components of the specimens was not obvious, and the specimens showed elastic responses. As the displacement of the frame-beam end increased, the concrete in the junction between the ring beam and the frame beam began to crack, and the specimens entered the crack-development stage. Subsequently, several staggered diagonal cracks appeared on the surface of the ring-beam joint and the frame beam close to the ring beam. However, when the load reached 85–90% of the peak bearing capacity, specimens S9 and S6 showed different failure trends from the other specimens. For specimens S9 and S6, the cracks in the joint area almost did not develop any further, while the cracks in the frame beam and at the junction between the frame beam and the ring beam developed rapidly with the crushing of the frame-beam concrete, as shown in Figure 4a,c. By contrast, for the other specimens (i.e., S1, S2, S3, S4, S5, S7, S8, S10, and S11), the number and width of the staggered diagonal cracks in the ring-beam joint increased evidently. Ultimately, specimens S9 and S6 were damaged by the failure of the frame beam and the destruction of the junction between the frame beam and the ring beam, respectively, whereas the other specimens were damaged by the shear failure in joint area. None of the columns presented distinct fractures on the tests. The typical failure modes are illustrated in Figure 4.

### 3.2. Hysteretic-Response Analysis

Figure 5 illustrates the *P–*Δ hysteresis curves of all the specimens, in which *P* represents the vertical load applied to the end of the RC-frame beam and Δ denotes the corresponding vertical displacement. Initially, all the specimens showed elastic responses, the load and displacements increased linearly in general, and the stiffness and strength were not degraded. Subsequently, diagonal cracks appeared on the ring-beam surface, and the reinforcements in the joints yielded. The *P–*Δ curves gradually deviated from linearity as the displacements increased, and the specimens entered the elastic–plastic stage. At this stage, a certain amount of residual deformation occurred. Subsequently, the concrete cracks in the joint area developed further, and slippage between the reinforcement and the concrete occurred, which may be the reason for the pinching effects of the hysteresis curves. After the specimens reached the peak bearing capacity, the stiffness and strength degraded significantly. The load reduced slowly with the rapid increase in the displacement, which indicated that the specimens exhibited good ductility and energy-dissipation ability. In other words, the specimens allowed the dissipation of a significant amount of mechanical energy into internal energy [34].

### 3.3. Skeleton-Curve Analysis

#### 3.3.1. Skeleton-Curve Features

Figure 6 shows the *P–*Δ skeleton curves of the specimens. They were obtained by connecting the peak points on the hysteresis curves. It can be seen that all the skeleton curves exhibited similar S shapes. Initially, the stress of all the specimens was small, and the skeleton curves increased approximately linearly. As the frame-beam-end displacement increased, the slopes of the skeleton curves gradually decreased. This phenomenon can be attributed to the cracking of the ring-beam concrete and the yield of the ring reinforcements in the ring-beam-joint area. Subsequently, most of the ring reinforcements and the ring-beam stirrups yielded, and significant slippage between the reinforcements and the concrete occurred in the joint area and at the junction of the ring beam and the frame beam. Ultimately, a gentle descending section can be observed in the skeleton curves.

#### 3.3.2. Effects of Reinforcement Ratio of Ring Beam

The effects of the $\rho_{r}$ on the *P–*Δ skeleton curves were not obvious in the elastic stage, as shown in Figure 6a. However, increasing the $\rho_{r}$ enhanced the peak bearing capacity and the corresponding peak displacement in elastic–plastic stage. This may be because a higher $\rho_{r}$ can provide a stronger confining effect on core concrete. In addition, with the increase in the $\rho_{r}$, the slopes of the descending section decreased. Taking specimens S1 and S3 as examples, the peak bearing capacity and corresponding peak displacement of specimen S1 were 32.42% and 20.00% higher than those of specimen S3, respectively. The ultimate displacement of the specimen S1 was 1.17 times that of specimen S3. This means the specimens with higher $\rho_{r}$ exhibited a larger bearing capacity and better ductility in an appropriate range.

#### 3.3.3. Effects of CFRP-Strip Spacing

As shown in Figure 6b, changing the CFRP-strip spacing had a marginal influence on the *P*–Δ skeleton curves. Several previous studies proved that the decreases in CFRP-strip spacing can significantly improve the bearing capacity of PCCC columns under axial, eccentric, and lower cyclic loading [26,29,32]. However, the columns of all the specimens in this experimental study were almost undamaged, which may have led to the phenomenon described above.

#### 3.3.4. Effects of Ring-Beam Width

Figure 6c illustrates the effects of the b on the skeleton curves. It is not difficult to observe that the b had little effect on the *P–*Δ skeleton curves in the elastic stage. Subsequently, the slopes of the skeleton curves increased as the b rose in the elastic–plastic stage. The peak bearing capacity also increased with the increase in the b, whereas the corresponding peak displacement did not exhibit the same regularity. The peak bearing capacities of specimens S8 and S3 were 30.13% and 15.16% greater than those of specimen S7, while the corresponding peak displacements of specimens S8 and S3 were 14.29% and 28.57% higher than those of specimen S7. This can be attributed to the increasingly prominent confining action of the ring beam on the core concrete as the b increased. It appears that increasing the confining effect improves the peak bearing capacity, while the corresponding peak displacement may be increased only within a specific ring-beam-width range.

#### 3.3.5. Effects of Axial Compression Ratio

The effects of n on the *P–*Δ skeleton curves were similar to those of the aforementioned parameters on the *P–*Δ skeleton curves in the elastic stage, as demonstrated in Figure 6d. In the elastic–plastic stage, the slopes of the skeleton curves increased as the n increased. With further increases in the displacement, specimen S11 reached the peak bearing capacity earlier than the S3. For instance, the peak bearing capacity of specimen S11 was 9.63% higher than that of specimen S3, while the corresponding peak displacement of specimen S11 was 22% lower than that of specimen S3. Moreover, the decline in the speed of specimen S11 in the descending section of the skeleton curve was greater than that of specimen S3. This was probably due to the fact that the specimens with higher n accelerated the development of concrete cracks in the joint area, leading to rapid strength degradation.

#### 3.3.6. Effects of Reinforcement Ratio of Frame Beam

The effects of the $\rho_{b}$ on the *P–*Δ skeleton curves are shown in Figure 6e. Clearly, the three curves were very close to each other during the whole loading process. The peak bearing capacity and the corresponding peak displacement displayed no obvious differences. This may have been due to the fact that in a specific $\rho_{b}$ range, changes in the $\rho_{b}$ only changed the failure modes of the specimens.

## 4. P–Δ Hysteresis Model of Exterior PCRB Joints

In this section, on the basis of the softening constitutive relation of the concrete, the P−Δ hysteresis model of the exterior PCRB joints, considering the effects of the structural parameters (i.e., $\rho_{r}$, *b*, *n* and $\rho_{b}$), is proposed and evaluated with the test data.

### 4.1. Basic Assumptions

(1)The frame-beam section conforms to the plane-section hypothesis.(2)The tensile action of the concrete is ignored.(3)The effects of ring-beam-core-concrete shrinkage and creep are neglected.(4)The constitutive relation of the reinforcement is a bi-linear model.(5)The constitutive relation of the concrete proposed by Zhang and Hsu [35] is adopted, and it can be expressed as Equation (1).

(1)$$\sigma_{d}=\zeta f_{c}{}^{\prime }\left[2\left(\frac{\varepsilon_{d}}{\zeta \varepsilon_{0}}\right)-{\left(\frac{\varepsilon_{d}}{\zeta \varepsilon_{0}}\right)}^{2}\right]\frac{\varepsilon_{d}}{\zeta \varepsilon_{0}}\leq 1$$(2)$$\zeta =\frac{5.8}{\sqrt{f_{c}{}^{\prime }}}\frac{1}{\sqrt{1+400\varepsilon_{r}}}\leq \frac{0.9}{\sqrt{1+400\varepsilon_{r}}}$$
where $\sigma_{d}$ is the concrete’s compressive stress in the strut (direction d), ζ is the softened coefficient, $f_{c}{}^{\prime }$ is the compressive strength of the cylinder, $f_{c}{}^{\prime }=0.79f_{\mathrm{cu}}$, and $f_{\mathrm{cu}}$ is the compressive strength of the cube, as shown in Table 2. The $\varepsilon_{d}$ and $\varepsilon_{r}$ are the concrete compressive strain in the strut (direction d) and the concrete tensile strain in the tie (direction r), respectively, as illustrated in Figure 7. Generally, the softened coefficient was limited to 0.9 when the $f_{c}{}^{\prime }$ was less than 42 MPa [36]. The $\varepsilon_{0}$ represents the corresponding peak strain of $f_{c}{}^{\prime }$, which can be determined by Equation (3) [36].
(3)$$\varepsilon_{0}=-0.002-0.001\left(\frac{f_{c}{}^{\prime }-20}{80}\right)$$

### 4.2. P–Δ Skeleton Curves

Clearly, the *P–*Δ skeleton curves of the exterior PCRB joints exhibited S shapes. To simplify the analysis, the yield point and peak point were selected as the turning points and the *P–*Δ skeleton curves were approximately simplified as three stages, namely the ascending stage (OA), the hardening stage (AB), and the descending stage (BC), as shown in Figure 8. A simplified tri-linear-relation model was proposed for predicting the *P–*Δ skeleton curves of the PCRB exterior joints.

#### 4.2.1. Yield-Bearing Capacity $P_{y}$ and Displacement $\Delta_{y}$

In this analysis, the frame-beam-end yield displacement $\Delta_{y}$ is composed of two parts, as follows:(4)$$\Delta_{y}=\delta_{y,b}+\delta_{y,j}$$
where $\delta_{y,b}$ and $\delta_{y,j}$ represent the frame-beam-end yield displacement caused by the frame-beam flexural deformation and shear deformation in the joint area, respectively.
(5)$$\delta_{y,b}=\theta_{y}L_{\mathrm{fb}}$$
(6)$$\theta_{y}=\varphi_{y}\frac{L_{\mathrm{fb}}}{3}+0.0025+a_{\mathrm{sl}}\frac{0.25\varepsilon_{y,\mathrm{fb}}d_{\mathrm{fb}}f_{y,\mathrm{fb}}}{\left(h_{b0}-a_{s}{}^{\prime }\right)\sqrt{{f^{\prime }}_{c}}}$$
where $\theta_{y}$ denotes the rotation angle of the frame beam when the joint yields [37]. The $L_{\mathrm{fb}}$ denotes the length of the frame beam. The $\varepsilon_{y,\mathrm{fb}}$ and $f_{y,\mathrm{fb}}$ are the yield strain and yield stress of the longitudinal reinforcement of the frame beam, respectively. The $d_{\mathrm{fb}}$ represents the diameter of the frame beam’s longitudinal reinforcement. The $a_{\mathrm{sl}}=1$ is the slippage coefficient. The $h_{b0}$ and $a_{s}{}^{\prime }$ denote the efficient height of the frame beam and the thickness of the protective concrete layer, respectively. The $f_{c}{}^{\prime }$ is the compressive strength of the cylinder. The $\varphi_{y}$ denotes the yield curvature of the frame-beam section, which can be obtained by analyzing the strain distribution of frame-beam section, as expressed in Equation (7).
(7)$$\varphi_{y}=\frac{\varepsilon_{y}}{\left(1-\xi_{y}\right)h_{b0}}$$
(8)$$\xi_{y}={\left(\eta^{2}A^{2}+2\eta B\right)}^{1/2}-\eta A$$
(9)$$A=\rho +\rho^{\prime }$$
(10)$$B=\rho +\left(a_{s}{}^{\prime }/h_{b0}\right)\rho^{\prime }$$
where $\xi_{y}$ is the relative-compression-zone height of the frame beam. The η stands for the ratio of the elastic modulus of the reinforcement to the concrete. The ρ and $\rho^{\prime }$ represent the longitudinal reinforcement ratios in the compression zone and in the tensile zone, respectively. The frame-beam longitudinal reinforcement was symmetrically arranged in this study and is denoted as $\rho =\rho^{\prime }$.

Based on the equilibrium condition of the force, the yield-bearing capacity $P_{y}$ can be expressed as Equation (11).
(11)$$P_{y}=\frac{M_{y}}{L_{\mathrm{fb}}}$$
(12)$$M_{y}=b_{b}h_{{}_{b0}}^{3}\varphi_{y}\left\{E_{c}\frac{\xi_{y}^{2}}{2}\left(0.5\left(1+\frac{a_{s}{}^{\prime }}{h_{b0}}\right)-\frac{\xi_{y}}{3}\right)+\frac{E_{s}}{2}\left[\left(1-\xi_{y}\right)\rho +\left(\xi_{y}-\frac{a_{s}{}^{\prime }}{h_{b0}}\right)\rho^{\prime }\right]\left(1-\frac{a_{s}{}^{\prime }}{h_{b0}}\right)\right\}$$
where $M_{y}$ is the moment of the frame beam upon the yield of the specimen [37]. The $E_{c}$ and $E_{s}$ are the elastic moduli of the concrete and the reinforcement, respectively.
(13)$$\delta_{y,j}=\gamma_{j}L_{\mathrm{fb}}$$
(14)$$\gamma_{j}=\delta_{j}/h_{b}$$
where $\gamma_{j}$ denotes the shear strain of the joint core. The $\delta_{j}$ is the shear deformation of the joint core, as shown below:(15)$$\delta_{j}=\frac{\delta_{\mathrm{str}}}{\cos\theta }=\frac{\varepsilon_{d}\sqrt{a_{s}^{2}+b_{s}^{2}}}{\cos\theta }$$
where $\delta_{\mathrm{str}}$ represents the deformation of the joint-core concrete in the strut. The compressive stress in the strut $\sigma_{d}$ is obtained by the following equation:(16)$$\sigma_{d}=N_{\mathrm{str}}/A_{\mathrm{str}}=V_{j}/A_{\mathrm{str}}\cos\theta$$
where $V_{j}$ denotes the shear force of the ring-beam joint when the specimen yields. The θ is the angle between the diagonal strut and the horizontal direction. The $A_{\mathrm{str}}$ represents the efficient area of the diagonal strut. The $a_{s}$ and $b_{s}$ are the height and width of the diagonal strut, respectively, which can be determined in accordance with Li [38]. Subsequently, according to the softening constitutive relation of the concrete, as expressed in Equation (1), the concrete’s compressive strain in the strut $\varepsilon_{d}$ can be obtained.

Ultimately, the yield-bearing capacity and yield displacement can be calculated by Equations (4) and (11), as listed in Table 3. The $P_{y}^{t}$ and $P_{y}^{c}$ are the experimental values and the calculated values of the yield-bearing capacity, respectively. It can be observed that the $P_{y}^{c}$ is significantly higher than the $P_{y}^{t}$. This is due to the fact that the frame beam’s longitudinal reinforcement inserted into the joint area yielded, while the longitudinal reinforcement of the frame beam close to the ring beam did not yield completely when joint-shear failure occurred.

However, according to Equation (11), the frame beam’s longitudinal reinforcement is considered to yield completely, which leads to larger calculated values. Therefore, based on the characteristic value of the longitudinal reinforcement ratio of the frame beam ($\lambda_{\mathrm{bs}}=\rho_{b}f_{y,\mathrm{bs}}/f_{\mathrm{cu}}$), a regression analysis of the test results was carried out, and a modified calculation formula for the yield-bearing capacity was proposed, as shown below:(17)$$P_{y}{}^{\prime }=\left(1.121-1.385\lambda_{\mathrm{bs}}\right)P_{y}$$
where $\Delta_{y}^{t}$ and $\Delta_{y}^{c}$ are the test values and the calculated values of the yield displacement of the specimens, respectively. Similarly, $\Delta_{y}^{c}$ is obviously lower than $\Delta_{y}^{t}$, owing to the fact that the confining effect of the ring reinforcement on the core concrete is ignored in the theoretical calculation. In fact, the specimens with higher $\rho_{r}$ exhibited greater deformability, as shown in Figure 6a. Therefore, a regression analysis of the test data was conducted based on the characteristic value of the ring-reinforcement ratio ($\lambda_{\mathrm{rs}}=\rho_{r}f_{y,\mathrm{rs}}/f_{\mathrm{cu}}$), and a modified calculation formula for the yield displacement is as follows:(18)$$\Delta_{y}{}^{\prime }=\left(1.442+1.034\lambda_{\mathrm{rs}}\right)\Delta_{y}$$

The calculated values obtained from Equations (17) and (18) are listed in Table 3. The results show that the modified predicted values agreed well with the test data.

#### 4.2.2. Peak Bearing Capacity $P_{m}$ and Displacement $\Delta_{m}$

The displacement $\Delta_{m}$ corresponding to the peak bearing capacity was calculated by the hardening-stage stiffness of the skeleton curves $k_{1}$, as expressed in Equation (19).
(19)$$\Delta_{m}=\frac{1}{k_{1}}\left(P_{m}-P_{y}\right)+\Delta_{y}$$
where $P_{m}$ is the peak bearing capacity, which can be calculated according to the formula suggested by Li [38]. Additionally, the test results indicated that $k_{1}$ is relevant to the characteristic value of the longitudinal reinforcement ratio of the frame beam $\lambda_{\mathrm{bs}}$, the characteristic value of the ring-reinforcement ratio of the ring beam $\lambda_{\mathrm{rs}}$, the influence coefficient of the joint dimension $k_{\mathrm{rb}}$, and the axial compression ratio *n*. Regarding the regression analysis of the test data in this paper, $k_{1}$ can be expressed as Equation (20).
(20)$$k_{1}=2.905+1.386\lambda_{\mathrm{rs}}-3.475k_{\mathrm{rb}}-1.686\lambda_{\mathrm{bs}}+0.329n$$
(21)$$k_{\mathrm{rb}}=\frac{D_{c}}{2b+D_{c}}$$
where $D_{c}$ denotes the diameter of the PCCC columns and b represents the width of the ring beam.

#### 4.2.3. Ultimate Bearing Capacity $P_{u}$ and Displacement $\Delta_{u}$

As illustrated in Figure 8, the ultimate bearing capacity $P_{u}$ is defined as 85% of the peak bearing capacity $P_{m}$. Therefore, the $P_{u}$ and the corresponding displacement $\Delta_{u}$ can be calculated as Equations (22) and (23).
(22)$$P_{u}=0.85P_{m}$$
(23)$$\Delta_{u}=\frac{1}{k_{2}}\left(P_{u}-P_{m}\right)+\Delta_{m}$$
(24)$$k_{2}=1.109-0.204\lambda_{\mathrm{rs}}-0.897k_{\mathrm{rb}}-3.246\lambda_{\mathrm{bs}}-0.153n$$
where $k_{2}$ denotes the descending stage stiffness of the skeleton curves, which was obtained through a regression analysis of the experimental data.

#### 4.2.4. Evaluation of the Proposed Model of Skeleton Curve

The test results and the calculated values of the important turning points of the skeleton curves are summarized in Table 4. Additionally, the comparisons between the experimental skeleton curves and the theoretical skeleton curves are illustrated in Figure 9. As can be seen from the comparisons, the theoretical skeleton curves were in good agreement with the experimental data, indicating that the model proposed above has high precision.

### 4.3. Hysteresis Rules

The model of the P−Δ skeleton curves of the exterior PCRB joints under low cyclic loading was described in Section 4.2. However, to establish a complete model for predicting P−Δ hysteresis curves, the hysteresis rules (i.e., the loading and unloading rules) also needed to be determined. Some existing models can provide references for the hysteresis rules of specimens [39,40,41]. To simply the analysis, on the basis of the Clough bi-linear degradation model [39], considering the effects of the different parameters (i.e., the $\rho_{r}$, b, n, and $\rho_{b}$) on the unloading stiffness $k_{u}$, the hysteresis rules of the exterior PCRB joints under low cyclic loading are proposed in this study. 

As shown in Section 3.2 and Section 3.3, the loading stiffness and unloading stiffness were close to elastic stiffness $k_{0}$($k_{0}=P_{y}/\Delta_{y}$) before the joints yielded, since only a few slight cracks occurred in the concrete surface. Subsequently, the unloading stiffness $k_{u}$, which is defined as the straight slopes when the peak load decreases to zero, degraded gradually with the increase in the frame-beam-end displacement after the joints yielded. The trend of the unloading stiffness degradation is illustrated in Figure 10, where Δ represents the displacement amplitude corresponding to the unloading point. It can be seen that $k_{u}/k_{0}$ exhibited the regulation of the power-function curve with the enhancement of the $\Delta /\Delta_{y}$. Hereby, the unloading stiffness $k_{u}$ can be fitted by using the regression analysis of the test data, as expressed below:(25)$$k_{u}=k_{0}a{\left(\frac{\Delta }{\Delta_{y}}\right)}^{c}$$
where *a* and *c* are the calculated coefficients. The specific expressions are as follows:(26)$$a=9.991-8.390\lambda_{\mathrm{rs}}-5.341k_{\mathrm{rb}}-9.204\lambda_{\mathrm{bs}}-1.469n$$
(27)$$c=-0.811+5.173n-1.130\lambda_{\mathrm{rs}}+0.221k_{\mathrm{rb}}-0.185\lambda_{\mathrm{bs}}$$

The unloading stiffness at various displacement amplitudes was calculated according to Equations (25)–(27), and the calculation results are summarized in Table 5. Additionally, the loading and unloading rules of the exterior PCRB joints under low cyclic loading are depicted in Figure 11. The detailed steps are as follows:
(a)Before the specimens yielded, the P−Δ skeleton curve was loaded and unloaded in the positive and negative directions with elastic stiffness $k_{0}$, as shown in Figure 11 (Point 0→1);(b)After the specimens yielded, the P−Δ skeleton curve was loaded in the positive direction (Point 1→2) and then unloaded from the displacement-amplitude point (Point 2) to P=0 point (Point 3), with the calculated unloading stiffness $k_{u}$; (c)When the reverse loading reached the yield load, the P=0 point of forward unloading (Point 3) pointed to the reverse yield point of the skeleton curve (Point 4). The reverse loading and unloading stiffness were the same as the forward loading and unloading stiffness (i.e., Point 4→5 and Point 5→6). Subsequently, the P=0 point of the reverse unloading (Point 6) pointed to the forward unloading point of the skeleton curve (Point 7). The duplicate process was followed based on the aforementioned rules. As shown in Figure 11, the traveling route of the model was determined according to the number, from small to large.

### 4.4. Verification of the Proposed Model of Hysteresis Curves

Using the skeleton-curve-prediction model and the hysteresis rules proposed above, the *P–*Δ hysteresis model of the exterior PCRB joints was established. The comparisons between the experimental hysteresis curves and the theoretical hysteresis curves are demonstrated in Figure 12. Most of the theoretical curves were in good agreement with the experimental curves, demonstrating that the established hysteresis model showed favorable precision. A small number of theoretical curves and experimental curves, such as S6, S10, and S11, had certain deviations, which may have been due to the hysteretic pinching caused by longitudinal-steel-bar slippage and crack propagation, were not considered.

## 5. Conclusions

Experimental investigations on 11 exterior PCRB joints under low cyclic loading were carried out. The effects of several structural variables on the seismic behaviors were analyzed. The conclusions can be summarized within the scope of the aforementioned parameters, as follows:
(1)Three failure modes of the exterior PCRB joints (i.e., the failure of the frame beam, the failure of the junction of the frame beam and the ring beam, and the shear failure of the ring-beam joint) were observed during the tests. The reinforcement ratio of the frame beam obviously affected the failure modes of the exterior PCRB joints.(2)With the increase in the ring-beam-reinforcement ratio, the axial compression ratio, and the ring-beam width, the pinching effect of the P−Δ hysteresis curves decreased, and the slope of the descending-stage skeleton curve decreased, whereas the frame-beam-reinforcement ratio and the CFRP-strip spacing had no distinct impact on the hysteresis curves, nor on the skeleton curves.(3)Based on the softening constitutive relation of the concrete, a simplified tri-linear-relation model for predicting the P−Δ skeleton curves of exterior PCRB joints was proposed, considering the effects of the reinforcement ratio of the ring beam, the axial compression ratio, the ring-beam width, and the reinforcement ratio of the frame beam. The predicted theoretical curves agreed well with the test data.(4)On the basis of the Clough bi-linear degradation model, the loading and unloading rules of the exterior PCRB joints under low cyclic loading were determined by the analysis of the P−Δ skeleton curves and the regression analysis of the experimental data. Subsequently, a model for estimating the P−Δ hysteresis curves of exterior PCRB joints was established using the P−Δ skeleton-curve-prediction model and the hysteresis rules. The established model of the P−Δ hysteresis curves has acceptable accuracy.(5)Although further research is needed, the proposed force–displacement hysteresis model is considered by the authors to be a useful tool for evaluating the hysteresis behavior of exterior PCRB joints under low cyclic loading.

## Figures and Tables

**Figure 1 polymers-15-02008-f001:**
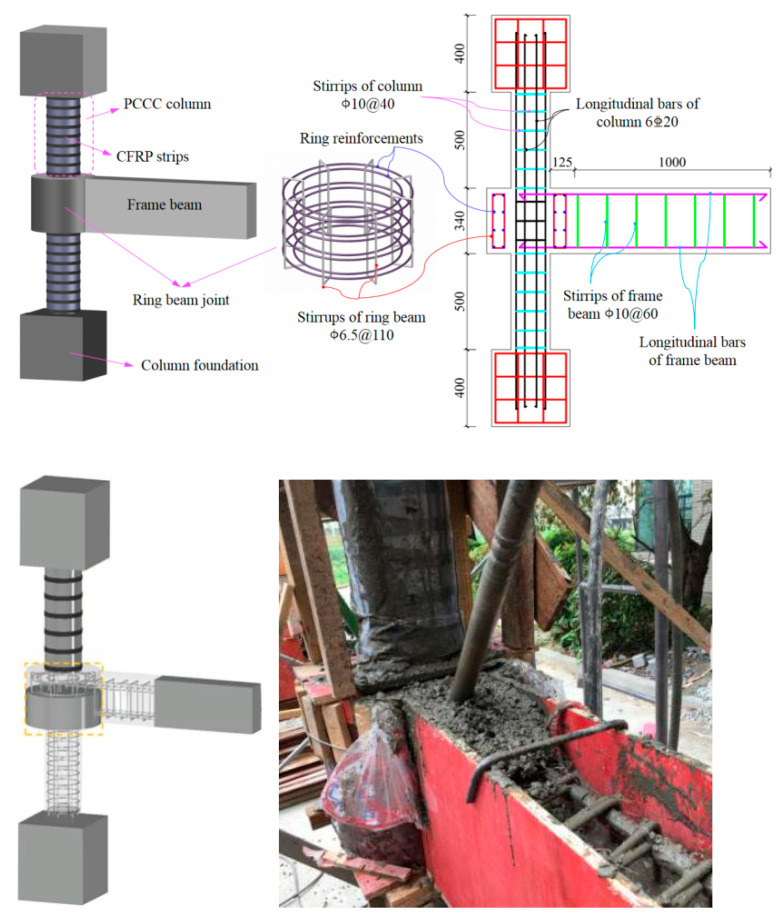
Specimen dimensions and reinforcement skeleton.

**Figure 2 polymers-15-02008-f002:**
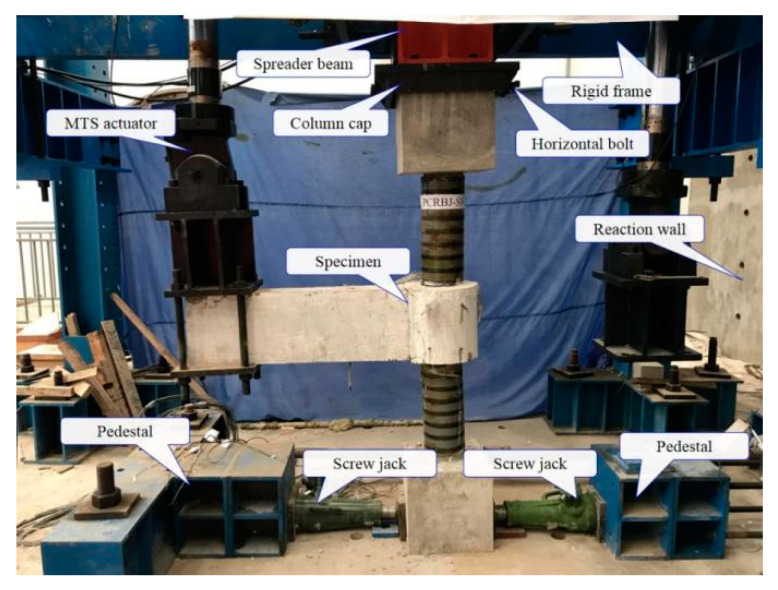
A general view of the test setup.

**Figure 3 polymers-15-02008-f003:**
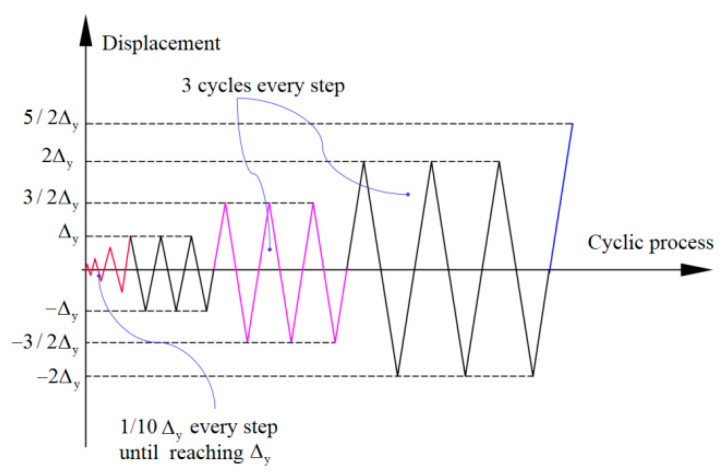
Imposed time-history displacement.

**Figure 4 polymers-15-02008-f004:**
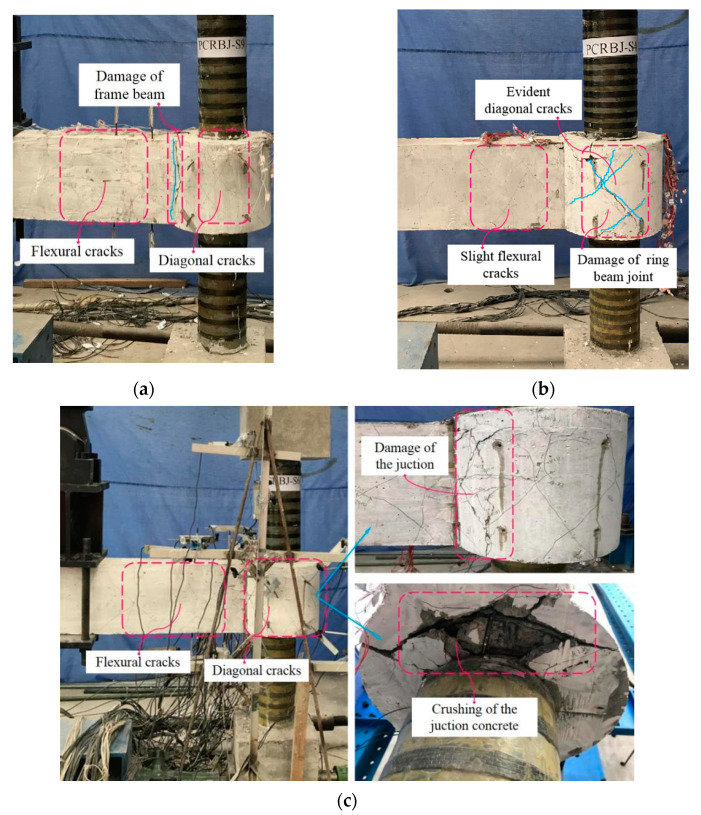
(**a**) Specimen S9; (**b**) Specimen S4; (**c**) Specimen S6. Typical failure modes of the specimens.

**Figure 5 polymers-15-02008-f005:**
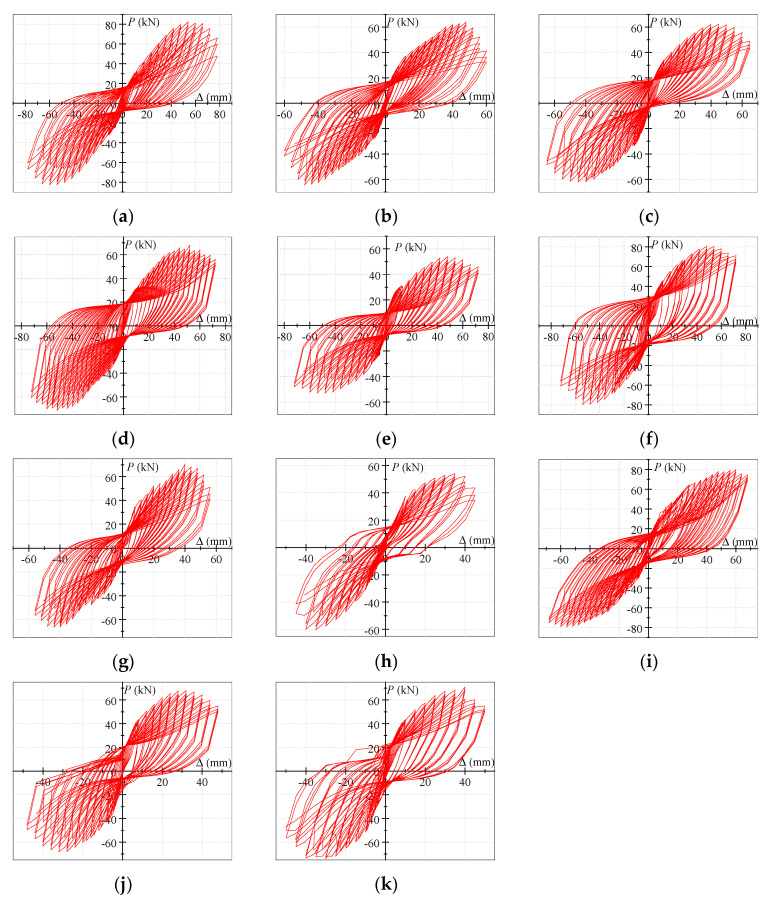
(**a**) S1; (**b**) S2; (**c**) S3; (**d**) S4; (**e**) S5; (**f**) S6; (**g**) S7; (**h**) S8; (**i**) S9; (**j**) S10; (**k**) S11. *P–*Δ hysteresis loops of the specimens.

**Figure 6 polymers-15-02008-f006:**
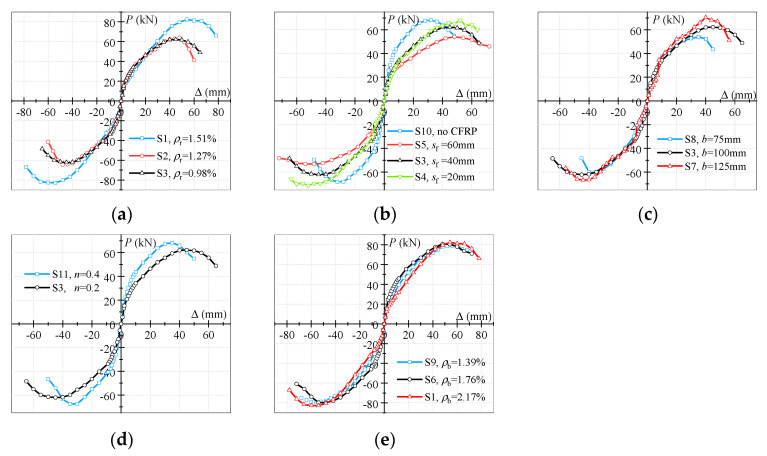
(**a**) Reinforcement ratio of ring beam; (**b**) CFRP-strip spacing; (**c**) ring-beam width; (**d**) axial compression ratio; (**e**) reinforcement ratio of frame beam. Effects of different parameters on the *P–*Δ skeleton curves.

**Figure 7 polymers-15-02008-f007:**
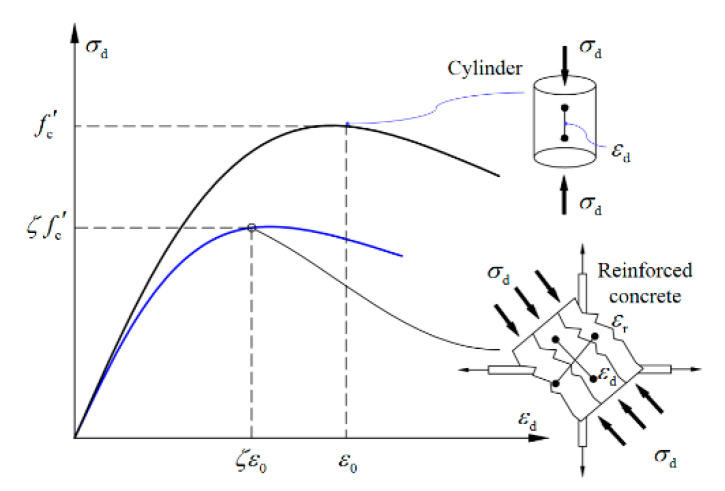
The softening constitutive relation of the concrete.

**Figure 8 polymers-15-02008-f008:**
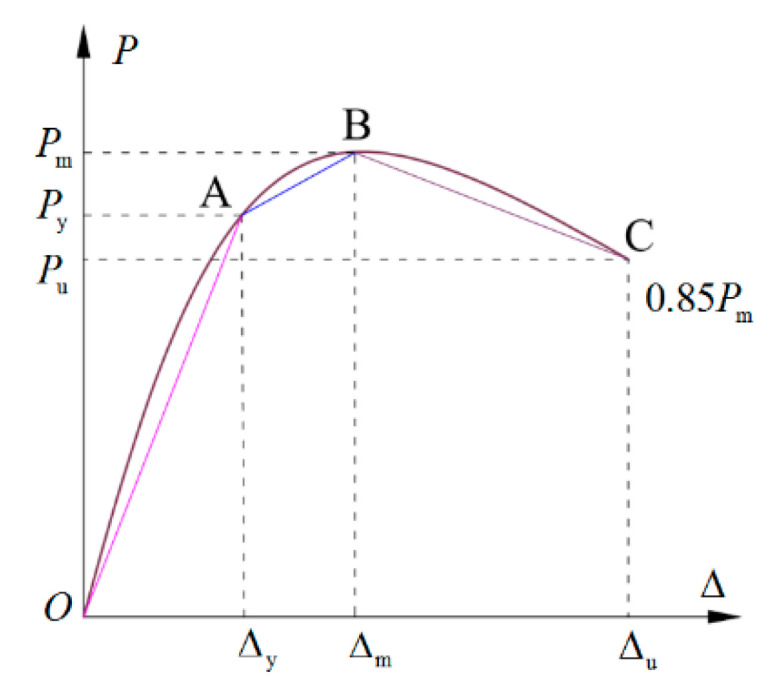
The key turning points of the skeleton P–Δ curves.

**Figure 9 polymers-15-02008-f009:**
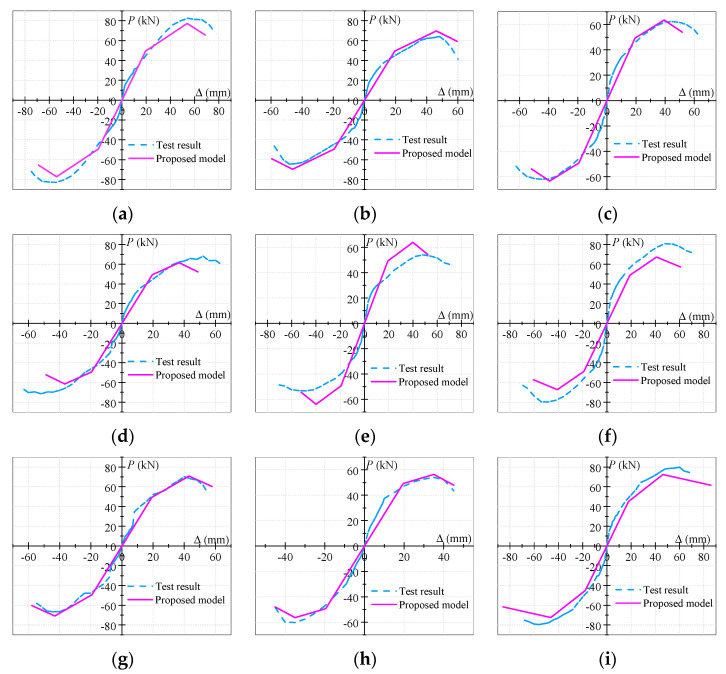
(**a**) S1; (**b**) S2; (**c**) S3; (**d**) S4; (**e**) S5; (**f**) S6; (**g**) S7; (**h**) S8; (**i**) S9; (**j**) S10; (**k**) S11. Comparisons between the experimental skeleton curves and theoretical skeleton curves.

**Figure 10 polymers-15-02008-f010:**
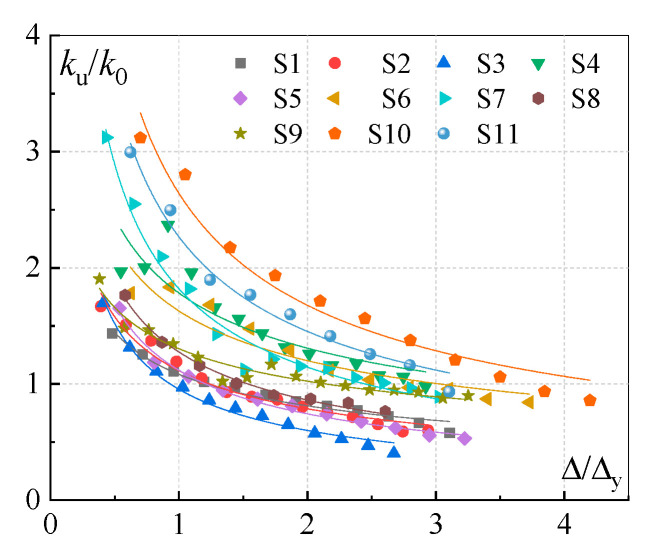
The trend of the unloading stiffness degradation.

**Figure 11 polymers-15-02008-f011:**
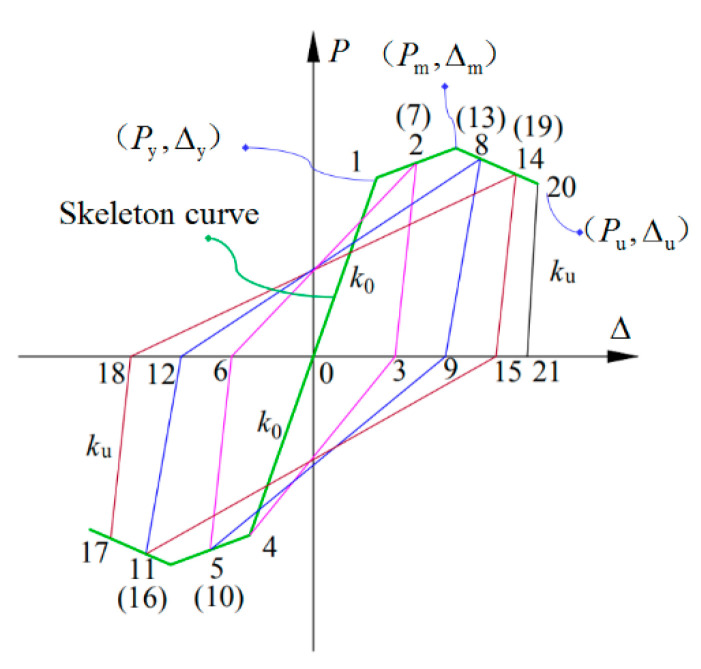
The loading and unloading rules of the exterior PCRB joints under low cyclic loading.

**Figure 12 polymers-15-02008-f012:**
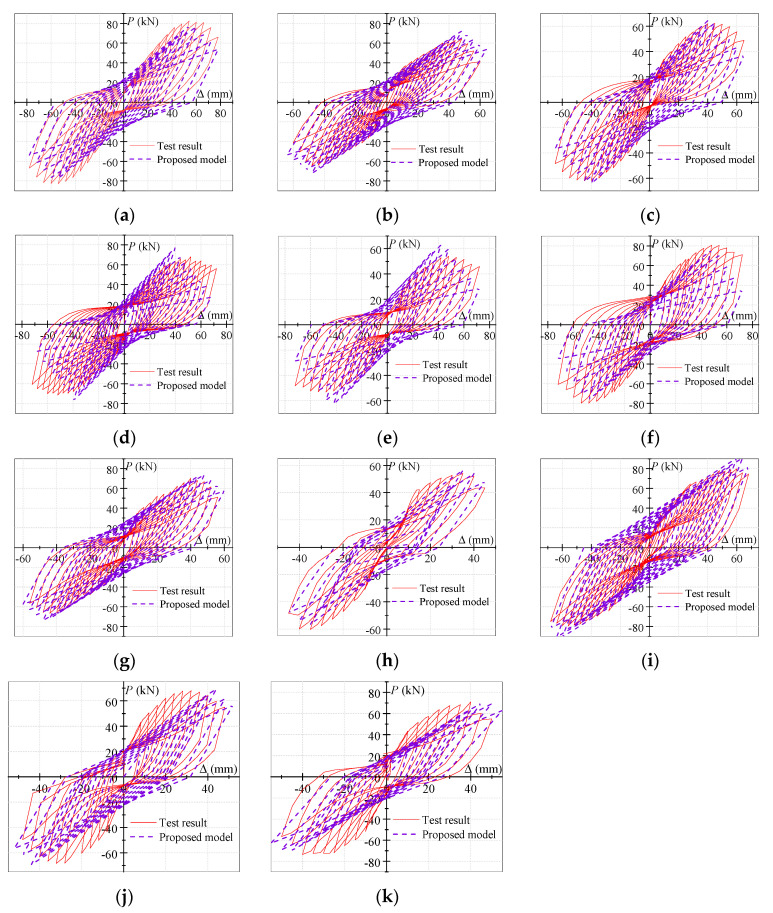
(**a**) S1; (**b**) S2; (**c**) S3; (**d**) S4; (**e**) S5; (**f**) S6; (**g**) S7; (**h**) S8; (**i**) S9; (**j**) S10; (**k**) S11. Comparisons between the experimental hysteresis curves and theoretical hysteresis curves.

**Table 1 polymers-15-02008-t001:** Detailed design parameters of the specimens.

Specimen	Ring Beam *b* × *h* (mm)	$\mathrm{CFRP}-\mathrm{Strip}\;\mathrm{Spacing}\;s_{f}^{}$ (mm)	$\mathrm{Reinforcement}\;\mathrm{Ratio}\;\mathrm{of}\;\mathrm{Ring}\;\mathrm{Beam}\;\rho_{r}$	$\mathrm{Reinforcement}\;\mathrm{Ratio}\;\mathrm{of}\;\mathrm{Frame}\;\mathrm{Beam}\;\rho_{b}$	Axial Compression Ratio *n*
S1	100 × 340	40	1.51% (4A10 + 4A8)	2.17% (4C20)	0.2
S2	100 × 340	40	1.27% (4A8 + 4A8)	2.17% (4C20)	0.2
S3	100 × 340	40	0.98% (4A8 + 4A6.5)	2.17% (4C20)	0.2
S4	100 × 340	20	0.98% (4A8 + 4A6.5)	2.17% (4C20)	0.2
S5	100 × 340	60	0.98% (4A8 + 4A6.5)	2.17% (4C20)	0.2
S6	100 × 340	40	1.51% (4A10 + 4A8)	1.76% (3C18)	0.2
S7	125 × 340	40	1.05% (4A10 + 4A6.5)	2.17% (4C20)	0.2
S8	75 × 340	40	1.04% (4A6.5 + 4A6.5)	2.17% (4C20)	0.2
S9	100 × 340	40	1.51% (4A10 + 4A8)	1.39% (3C16)	0.2
S10	100 × 340	-	0.98% (4A8 + 4A6.5)	2.17% (4C20)	0.2
S11	100 × 340	40	0.98% (4A8 + 4A6.5)	2.17% (4C20)	0.4

Notes: “4A10 + 4A8” means there are 4 ring reinforcements with a diameter of 10 mm and 4 ring reinforcements with a diameter of 8 mm in the ring-beam zone. The number before the “+” represents the outer-ring reinforcement, and the number after the “+” represents the inner-ring reinforcement. (a) Reinforcement ratio of ring beam $\rho_{r}=A_{\mathrm{rs}}/bh$; $A_{\mathrm{rs}}$ is the total area of ring reinforcement in the ring beam, *b* and *h* are the width and height of ring beam, respectively. (b) Reinforcement ratio of frame beam $\rho_{b}=A_{\mathrm{bs}}/b_{b}h_{b}$: $A_{\mathrm{bs}}$ is the total area of longitudinal reinforcement in the frame beam, $b_{b}$ and $h_{b}$ are the width and height of frame beam, respectively. (c) The $s_{f}$ represents the spacing of the CFRP strips, where the spacing is the edge-to-edge distance between the adjacent CFRP strips.

**Table 2 polymers-15-02008-t002:** Mechanical properties of the materials used in the test.

Materials	Yield Strength (MPa)	Ultimate Strength (MPa)	Elastic Modulus (MPa)
CFRP (Tension)	-	3796	274,000
PVC (Tension)	-	69.6	2590
Concrete (Compression)	-	22.9	31,200
Reinforcement (Tension) A6.5	323	542	197,000
Reinforcement (Tension) A8	308	426	201,000
Reinforcement (Tension) A10	313	432	197,000
Reinforcement (Tension) C16	451	620	195,000
Reinforcement (Tension) C18	465	633	195,000
Reinforcement (Tension) C20	446	611	199,000

**Table 3 polymers-15-02008-t003:** Comparison between the calculated values and the test results.

Specimen	$\;P_{y}^{t}$ (kN)	${\;\Delta }_{y}^{t}$(mm)	$P_{y}^{c}$ (kN)		$\Delta_{y}^{c}$ (mm)		$P_{y}^{t}/P_{y}^{c}$		$\Delta_{y}^{t}{/\Delta }_{y}^{c}$	
			Equation (11)	Equation (17)	Equation (4)	Equation (18)	Equation (11)	Equation (17)	Equation (4)	Equation (18)
S1	57.5	25.1	92.1	49.3	11.7	19.5	0.624	1.166	2.145	1.287
S2	45.8	20.4	92.1	49.3	11.9	19.4	0.497	0.929	1.714	1.052
S3	51.4	24.3	92.1	49.3	12.2	19.3	0.558	1.043	1.992	1.259
S4	47.7	21.9	92.1	49.3	12.3	19.5	0.518	0.968	1.780	1.123
S5	40.5	22.4	92.1	49.3	12.2	19.3	0.440	0.822	1.836	1.161
S6	53.5	19.3	75.5	48.8	12.0	18.9	0.709	1.096	1.608	1.021
S7	49.0	18.6	92.1	49.3	12.1	19.2	0.532	0.994	1.537	0.969
S8	44.6	17.3	92.1	49.3	12.4	19.5	0.484	0.905	1.395	0.887
S9	50.5	20.9	60.4	45.1	11.3	17.8	0.836	1.120	1.850	1.174
S10	49.9	11.5	92.1	49.3	12.1	19.2	0.542	1.012	0.950	0.599
S11	53.6	16.1	92.1	49.3	11.9	18.9	0.582	1.087	1.353	0.852
Average value	-	-	-	-	-	-	0.575	1.013	1.651	1.035
Standard deviation	-	-	-	-	-	-	0.113	0.103	0.334	0.202

**Table 4 polymers-15-02008-t004:** The test results and calculated values of the key turning points of skeleton curves.

Specimen	Yield Point				Peak Point				Ultimate Point			
	$P_{y}^{t}$	$\Delta_{y}^{t}$	$P_{y}^{c}$	$\Delta_{y}^{c}$	$P_{m}^{t}$	$\Delta_{m}^{t}$	$P_{m}^{c}$	$\Delta_{m}^{c}$	$P_{u}^{t}$	$\Delta_{u}^{t}$	$P_{u}^{c}$	$\Delta_{u}^{c}$
S1	57.5	25.1	49.3	19.5	82.5	54.0	77.2	53.9	70.5	75.5	65.6	68.7
S2	45.8	20.4	49.3	19.4	64.1	48.1	69.7	46.1	54.6	54.8	59.2	59.5
S3	51.4	24.3	49.3	19.3	62.3	45.0	63.5	39.4	49.7	64.2	54.0	51.8
S4	47.7	21.9	49.3	19.5	68.1	52.1	61.5	36.7	59.9	63.9	52.3	48.7
S5	40.5	22.4	49.3	19.3	54.0	48.1	63.9	39.9	46.0	72.1	54.3	52.3
S6	53.5	19.3	48.8	18.9	80.8	48.0	67.3	40.9	72.1	71.1	57.2	60.7
S7	49.0	18.6	49.3	19.2	70.4	40.0	70.9	43.1	59.7	53.0	60.3	57.9
S8	44.6	17.3	49.3	19.5	54.1	35.0	56.4	35.0	45.9	43.8	47.9	45.1
S9	50.5	20.9	45.1	16.6	78.7	52.0	72.5	46.3	68.1	74.6	61.6	85.7
S10	49.9	11.5	49.3	20.0	68.0	32.1	64.5	40.6	58.0	45.5	54.8	53.2
S11	53.6	16.1	49.3	20.1	68.3	35.1	69.5	45.0	58.0	46.8	59.1	58.0

**Table 5 polymers-15-02008-t005:** The calculated unloading stiffnesses of the specimens.

Specimen	Variations in Unloading Stiffnesses of Specimens														
S1	$\Delta /\Delta_{y}$	0.48	0.72	0.96	1.20	1.44	1.68	1.91	2.15	2.39	2.63	2.87	3.11	-	-
	$k_{u}/k_{0}$	1.62	1.58	1.53	1.39	1.31	1.25	1.18	1.17	1.11	1.09	1.04	0.99	-	-
S2	$\Delta /\Delta_{y}$	0.39	0.59	0.79	0.98	1.18	1.37	1.57	1.77	1.96	2.16	2.36	2.55	2.75	2.94
	$k_{u}/k_{0}$	2.26	1.92	2.08	1.74	1.57	1.40	1.29	1.24	1.18	1.16	1.19	1.15	1.16	1.05
S3	$\Delta /\Delta_{y}$	0.41	0.62	0.82	1.03	1.24	1.44	1.65	1.85	2.06	2.27	2.47	2.67	-	-
	$k_{u}/k_{0}$	3.40	2.57	2.10	1.81	1.68	1.50	1.40	1.37	1.33	1.36	1.31	1.33	-	-
S4	$\Delta /\Delta_{y}$	0.37	0.55	0.73	0.91	1.28	1.46	1.65	1.82	2.01	2.19	2.38	2.55	2.74	2.92
	$k_{u}/k_{0}$	4.02	3.98	4.73	3.93	3.14	2.92	2.67	2.68	2.57	2.50	2.66	2.45	2.89	3.42
S5	$\Delta /\Delta_{y}$	0.54	0.81	1.07	1.34	1.61	1.88	2.15	2.41	2.68	2.95	3.22	-	-	-
	$k_{u}/k_{0}$	2.30	1.73	1.51	1.36	1.39	1.30	1.29	1.23	1.16	1.11	1.06	-	-	-
S6	$\Delta /\Delta_{y}$	0.62	0.93	1.24	1.56	1.87	2.18	2.49	2.80	3.10	3.41	3.74	-	-	-
	$k_{u}/k_{0}$	8.12	4.62	2.91	2.08	1.77	1.80	1.73	1.77	1.77	2.02	2.13	-	-	-
S7	$\Delta /\Delta_{y}$	0.43	0.65	0.86	1.08	1.29	1.51	1.72	1.94	2.15	2.37	2.58	2.80	3.02	-
	$k_{u}/k_{0}$	2.63	2.23	1.96	2.17	1.87	1.57	1.44	1.33	1.27	1.15	1.11	1.05	0.88	-
S8	$\Delta /\Delta_{y}$	0.58	0.87	1.16	1.45	1.74	2.02	2.32	2.60	-	-	-	-	-	-
	$k_{u}/k_{0}$	1.84	1.38	1.13	0.96	0.84	0.85	0.83	0.71	-	-	-	-	-	-
S9	$\Delta /\Delta_{y}$	0.38	0.77	0.96	1.15	1.34	1.53	1.72	1.92	2.11	2.49	2.68	2.87	3.06	3.26
	$k_{u}/k_{0}$	3.03	2.59	4.43	3.20	2.68	2.20	1.90	1.73	1.57	1.39	1.28	1.26	1.24	1.16
S10	$\Delta /\Delta_{y}$	0.70	1.05	1.39	1.74	2.09	2.44	2.79	3.14	3.49	3.83	4.18	-	-	-
	$k_{u}/k_{0}$	2.52	1.60	1.24	1.04	0.93	0.93	0.86	0.78	0.68	0.61	0.55	-	-	-
S11	$\Delta /\Delta_{y}$	0.62	0.93	1.24	1.55	1.87	2.18	2.49	2.80	3.10	-	-	-	-	-
	$k_{u}/k_{0}$	2.40	1.84	1.35	1.17	1.04	0.96	1.00	1.06	0.89	-	-	-	-	-

## Data Availability

Some or all data, models, or code that support the findings of this study are available from the corresponding author upon reasonable request (Table 2, Table 3, Table 4 and Table 5).

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
