# Peer review of "Force–Displacement Hysteresis Model of Exterior PCRB Joints under Low Cyclic Loading"

_polymers, 2023, doi:10.3390/polym15092008_

Round 1

Reviewer 1 Report

The paper presents a topic of current interest but the authors need to address all the issues highlighted in the attached report. After that, the reviewer will evaluate the suitability of the work.

Author Response

The authors would like to thank the reviewers for the positive and constructive comments. The manuscript has been revised accordingly. Here, we would like to submit our replies to the reviewers and the revised manuscript. We hope the manuscript could meet the reviewers’ requirements. We have highlighted the changes in the revised manuscript by red coloured text.

Reviewer 2 Report

The authors have done a comprehensive study. I believe that if accepted, it will contribute to the literature.

According to which standard did you determine the frame size?

Longitudinal Young's modulus of the CFRP material is given in Table 2. Did the loads on this material act in the longitudinal direction during the experiment?

A reference sample with steel stirrups could have been made for better understanding of the results.

Please expand the literature search.

Author Response

(The authors gave the same response as above.)

Round 2

Reviewer 1 Report

In the references section there are mistakes: wrong author name. It happens, for example, in references 35 and 41. It is necessary to address such issues. I will evaluate the paper suitability after that.

Author Response

Thanks for pointing out the error. We have made the corresponding modifications in the revised manuscript.

  1. Vaiana N.; Capuano R.; Rosati L. Evaluation of path-dependent work and internal energy change for hysteretic mechanical systems. Mechanical Systems and Signal Processing, 2023, 186: 109862.
  2. Capuano R.; Vaiana N.; Pellecchia D.; Rosati L. A solution algorithm for a modified Bouc-Wen model capable of simulating cyclic softening and pinching phenomena. IFAC-Papers On Line, 2022, 55 (20): 319-324.

Reviewer 2 Report

The manuscript can be accept.

Author Response

Thanks again for your useful and reasonable suggestions.

Round 3

Reviewer 1 Report

The authors addressed all the issues thus the manuscript can be accepted for publication.